# A Hybrid SEM-Neural Network Modeling of Quality of M-Commerce Services under the Impact of the COVID-19 Pandemic

Anca Mehedintu ![ID] and Georgeta Soava *![ID]

Department of Statistics and Economic Informatics, University of Craiova, A.I. Cuza 13, 200585 Craiova, Romania
* Correspondence: georgeta.soava@edu.ucv.ro; Tel.: +40-251-411317

**Abstract:** The research purpose is to contribute to the understanding of the COVID-19 pandemic impact on the intensification of commercial transactions on the mobile channel (m-commerce) and to identify the most significant factors that act on consumer behavior based on the development of a conceptual model to establish the influence of m-commerce service quality on customer satisfaction and loyalty. The data were collected through a survey addressed to customers who, during 2021–2022, made at least one purchase through m-commerce. The analysis was performed with SPSS Statistics and Amos software, using a hybrid approach: Structural Equation Modeling (SEM) and Artificial Neural Network (ANN). The research results confirm the hypotheses presented in this study. Both models identified the quality of services offered by m-commerce, satisfaction, and trust as determining factors for increasing consumer loyalty in virtual commerce. The novelty of this study consists of an interconnected analysis model of some variables specific to mobile commerce, which have not been used in this combination in the specialized literature. This research can be the basis of other research studies. In addition, it provides valuable results for the business environment (forecasts) and customers by obtaining improved, personalized, and secure commerce services.

**Keywords:** COVID-19; m-commerce services; customer satisfaction; customer loyalty; structural equations modeling; artificial neural network





## 1. Introduction

The COVID-19 pandemic has changed the world, dramatically affecting the activity of companies and consumers [1] and causing profound changes in how buyers interact with sellers [2]. Physical distancing has forced companies to adapt their way of relating to customers by redefining business processes [3]. Regardless of the degree of development of the countries' economies, online shopping has become essential [4]. Consumers have been forced to change their purchasing behavior from physical stores to digital platforms in the short or perhaps long term [2,5,6]. The consumer can be considered a person or a group that intends to order or use goods, products, or services purchased primarily for personal, social, family, household, and similar needs, which are not directly related to entrepreneurial or business activities [6]. Thus, e-commerce has seen a new impetus, becoming a highly used channel that has increased its popularity [7,8] and companies will have to manage it properly and integrate it with a variety of other online and offline channels in the future [9], and, at the same time, identify the limits of e-commerce in the context of consumer protection [10].

From the beginning of the 21st century until 2020, e-commerce has developed steadily; the economic crises of 2000 and 2008 affected the e-economy domain the least, and after two–three years, the situation returned to pre-crisis conditions [11]. Once the pace of EC development stabilized, and the device used has become the distinguishing criterion: traditional e-commerce is associated with the use of a laptop or desktop, and m-commerce with the use of mobile devices. In the digital economy, e-commerce has become an essential

tool, stimulating the competitiveness and innovation of companies in terms of how they promote their products and services while producing a transformation in people's behavior and creating a new consumer profile [12].

Electronic commerce through digital technologies ensures the direct connection of business partners, increasing the attractiveness of the business environment but also new opportunities for consumers [13]. E-commerce represents a fundamental condition for business development and establishing optimal relationships with customers [14]. E-commerce can be defined as the economic activity of buying and selling products and services through online platforms [15].

Mobile commerce (m-commerce) refers to the use of portable wireless devices to conduct online transactions (buying and selling goods and services) [16,17].

In the context of the COVID-19 crisis, existing business models had to, in addition to maintaining traditional methods, change their strategy by turning to e-commerce platforms. Both quarantine and social isolation were key mental states that led more people to shop online, with online sales up 47% year-on-year [8].

One of the main aspects that supported the exponential growth of online product purchases was the need to ensure individual safety against coronavirus [18].

Isolation and social distancing have reduced people's mobility in retail which has led to a shrinking of the market for non-essential products that are durable and 'put-away' [19]. Thus, the online consumption of goods and services increased while offline consumption decreased during the COVID-19 pandemic [20]. If, before the pandemic, online purchases were predominantly focused on electronic products and services and fashion products, during the pandemic they concentrated on health-related products, products necessary for working from home, and food products [21].

The most sought-after products were medicine, toiletries, disposable gloves, food and drink (e.g., canned goods), household appliances, toys, puzzles, picture books, exercise equipment, high-tech equipment, and articles of clothing [8,22]. At the same time, during the pandemic, and especially during the lockdown, there was a general increase in the online trade of food products [23,24], often due to the lack of alternatives, but is expected that the growth trend for the e-food sector will also be preserved in the post-COVID-19 period [25].

The main factors that influence consumer loyalty for e-commerce are the trust in the food products offered, the perceived value, and the attitude of the sellers towards their needs [26]. The pandemic has brought drastic changes to consumers' planned and unplanned buying habits [2], changed the traditional schemas used by consumers to evaluate offers and make a choice, and affected the traditional way in which consumers relate to retailers [27].

The studies carried out on customer satisfaction regarding the use of e-commerce during the pandemic show that they are satisfied and will continue to look for this type of commerce in the future and the long term, which will produce a major change in technological, social, and economic transformation [28]. Even after COVID-19, significant changes in daily life and shopping behavior will remain, which amplifies the need for research on consumer behavior to understand what factors influence the use of e-commerce [29,30].

Although at the level of all states, in the last decades there have been continuous increases in electronic commerce (e-commerce) and especially in mobile commerce (m-commerce), which amplified during the COVID-19 pandemic. This fact has caused companies to become aware of the potential of this commercial channel [21]. In 2021, Romania recorded electronic sales worth EUR 14 billion, which represents almost half of the total achieved in Eastern Europe [31].

Consumers have become aware of the conveniences offered by using e-commerce in general [32] and mobile commerce in particular [21]. The benefits of m-commerce (availability, purchasing facilities in terms of time and space) give consumers a more user-friendly experience than e-commerce. Thus, customers began to orientate more and more

towards this channel, and most e-commerce platforms adapted their content to be easier to use on mobile devices [14].

Although several studies have attempted to identify the determinants [33] that make consumers turn to m-commerce [34], there are also several uncertainties regarding the motivation of consumers to continue using m-commerce [35,36] and its development follows different trajectories depending on the country [37]. In this sense, m-commerce has a major influence on how companies interact with consumers, which makes the study of the quality of mobile shopping services extremely important. Therefore, the first objective of this study is to identify the significant factors that can lead to an increase in the quality of m-commerce services offered to consumers in Romania, thus influencing them to use the online purchase again.

After researching the specialized literature, I found that several studies analyze various models regarding the adoption of m-commerce [5,7,38]. At the same time, recent research focuses on the analysis of the impact of service quality on the future use of m-commerce and consumer loyalty in various countries or regions [35,39–43]. To ensure customer loyalty, m-commerce providers need to know the main factors that influence consumer satisfaction. Thus, another objective of this study is to build a conceptual model to identify the impact of the quality of electronic services on the behavior of m-commerce consumers in the example of Romanian consumers.

For the research, I used a hybrid approach, Structural Equation Modeling (SEM) and Artificial Neural Network (ANN), using SEM to test hypotheses regarding customer satisfaction with respect to the quality of m-commerce services and neural network analysis to identify the classification of significant determinants established in SEM, depending on their influence.

The results show that both models have identified that m-commerce service quality, satisfaction, and trust are the most important factors for increasing consumer loyalty in m-commerce. A significant result of the study is that by ANN, one can identify the degree to which the model's factors compete to increase the quality of m-commerce services and companies can take timely action to improve them.

In addition to identifying the factors that influence consumers' use of m-commerce, this paper contributes significantly to existing research on m-commerce adoption by developing a conceptual model on the impact of service quality on m-commerce consumer behavior. The conclusions of this study are significant for the business environment because companies can better understand and forecast consumer behavior, being able to organize their operations and marketing campaigns in such a way as to address the main identified factors, "satisfaction" and "trust", for increasing customer loyalty. At the same time, this study can bring advantages to customers through the possibility of offering improved, personalized, and high-security m-commerce services and can be the basis of other research studies.

Compared to the studies in the specialized literature, this work offers original contributions through the conceptual model created to examine the impact of the COVID-19 epidemic on the increase in the quality of m-commerce services and also through the use of neural networks to determine the most significant factors that compete for the increase in customer loyalty.

The research was structured in five sections. Section 2 discusses the theoretical background and identifies the hypothesis on which the research model developed. Section 3 describes the sample and presents the research methodology used. Section 4 presents the results and their interpretations. The final section highlights the conclusions, explores the main theoretical and managerial implications of the research, and highlights the study's main limitations and some possible future research directions.

## 2. Theoretical Background and Hypotheses Development

As a result of the COVID-19 pandemic, digital interactions with customers (sales, customer service, etc.) [43,44] have grown at a rapid pace, changing the outlook for e-

commerce development, with the number of e-commerce buyers exceeding 3.4 billion [45]. According to statistics, by 2040, almost 95% of all purchases will be made through electronic commerce [46]. As for Romania, e-commerce activity reached the threshold of EUR 5.6 billion at the end of 2020, 30% more than in 2019, and at the end of 2021, it reached the threshold of EUR 6.2 billion, almost 10% more than in 2020 [47].

Given the widespread appeal of mobile devices and the constant development of wireless networks, m-commerce is rapidly growing in popularity and becoming essential in the daily life [48]. The mobile commerce channel (m-commerce) represents the future of online markets [49]. Through easy accessibility to customers, m-commerce applications extend the time available for searching [50].

The COVID-19 pandemic has changed the behavior towards virtual commerce [51,52], with customers increasingly migrating to m-commerce due to a multitude of factors [53,54]. Thus, for companies' m-commerce channel transaction strategies to be successfully leveraged, they need to understand how to target customers' perceptions of m-commerce and identify the factors that affect its use [55]. Customers' shopping and payment preferences have changed [56]. They choose to buy while considering various factors: brand reputation, product quality, diversity, service quality, security, ease of use, and price competitiveness [27].

*2.1. M-Commerce Apps*

With the outbreak of the COVID-19 crisis, the purchasing behavior of consumers has changed and tend to rely more frequently on mobile applications [5,7,39]. It has led companies to prioritize their e-commerce transition to mobile platforms and adapt their websites for mobile commerce channel [57,58].

The creation of m-commerce websites has simplified the shopping experience [59] but also enhanced it by allowing consumers to browse multiple stores, obtain product information, and have the availability of purchasing from anywhere and at any time [60]. It is essential to ensure the quality of commercial applications from the user perspective (content, search, and navigation) to increase their credibility [61,62].

However, several problems may arise regarding the acceptance, adoption, and diffusion of m-commerce [63]. In this sense, the creation of successful m-commerce websites must start from the basic requirements for commercial smartphone applications [64] and provide benefits to users [65] while also considering their continuous improvement (increasing internet speed, expanding 5G and Wi-Fi networks, and increase accessibility and trust in mobile devices and applications) [66].

Customers rate the user experience of a website to assess the overall quality of an online store's services. Users must perceive ease of use [67] and simple access [68].

The quality of e-services can be measured by website design, customer support services, security/privacy, and fulfillment [69]. At the same time, the quality of services offered by the website can be characterized by six elements: appearance, content, organization, interaction, focus on the client, and insurance [8].

For m-commerce applications, the content of the displayed information must attract users' attention and be accessible from various devices [70,71]. Good organization of mobile applications can lead to increased quality and performance levels and provides users with easy and hassle-free access to application content [72].

The attractive appearance of an m-commerce application is one of the most important ways to attract new customers in the future [8,41,67]. Website design refers to all elements of the customer experience related to the website, including information quality, website aesthetics, purchase process, website convenience, product selection, price offers, website customization web, and system availability. Customers quantify m-commerce application design through aesthetic preferences [73] and information updates [8,67]. In this context, website design influences the perceived ease of use and influences customers' attitudes and behavioral intentions, positively affecting the quality of electronic services [40]. At the same time, using the mobile commerce channel implies the need to adapt e-commerce applications for compatibility with mobile devices to fit their smaller screens [74].

Consumers tend to perceive visual appeal first. Such a perception can induce positive attitudes, making them effectively search, browse, and evaluate the products they want and need [41].

Usage behavior and customer satisfaction in m-commerce applications depend on customer trust in m-banking services [75–77]. Several types of research have been dedicated to identifying the factors that influence users to adopt and use m-banking [78–81] and their impact on customer satisfaction [79].

Appropriately designed commerce applications will help users find what they are looking for and successfully make transactions through mobile phones [70].

Studies indicate that perceived usefulness and ease of use, information and service quality, user interface, perceived risk, and security all have direct consequences on user attitude, satisfaction, and loyalty [58].

Based on the specialized literature that analyzed the factors that make the use of the Web a compelling experience for its users, we formulated the following hypothesis:

**Hypothesis 1 (H1).** *Mobile site design and content (DC) has a positive and significant effect on m-commerce service quality (MQ).*

### 2.2. Customer Support for the M-Commerce

Several studies consider portability to be the greatest advantage of virtual commerce, as it allows consumers to access product information at anytime and anywhere, and to contact human or intelligent agents at any time, so that they can be properly informed [30,74]. With the mobile device, users can browse the products and can receive useful tips, contact details, information related to returns, and guarantees [30]. Customers are supported throughout the purchasing process (search, choice, purchase, and after-sales service) by the interactivity that suppliers offer them (active control, personalization, ubiquitous connectivity, connection, responsiveness, and synchronicity) [82]. Mobile devices have turned into personal assistants, helping customers access new information and instantly purchase digital content and various products and services [83]. As consumer attitudes in the online space are constantly changing, digital content is visited more carefully [84], and customers become more interactive in communication, seeking information to their questions at a certain time and place exactly as they wish, and spend much less time on transactions. Consumers have changed their passive role to an active one, which forces e-service providers to be proactive [85].

Based on these findings, the following hypotheses are formulated:

**Hypothesis 2 (H2).** *Customer support (CP) has a positive and significant effect on m-commerce service quality (MQ).*

### 2.3. Security and Privacy in M-Commerce

Customer trust in m-commerce is determined by ensuring the security of purchases (implementation of procedures to ensure the transfer of information and the security of credit card payments) and the protection of confidentiality (confidentiality of shared information, i.e., not to use the buyer's data for any purpose other than the one agreed with the buyer) [69,86].

Customers are always concerned about whether the website where they want to transact will protect them against fraud, so security has an important influence on the quality of m-commerce services [40]. Studies show that companies that carry out m-commerce activity are also very concerned about ensuring confidentiality when customers are not ready to disclose private data [68,87]. Thus, several factors have been identified that can influence the decisions of potential consumers to make their data available [73,87,88].

Considering the significant effect of security and privacy in the context of m-commerce, the following hypothesis is proposed:

**Hypothesis 3 (H3).** *Security/Privacy (SP) has a positive and significant effect on m-commerce service quality (MQ).*

*2.4. Fulfillment of the M-Commerce*

Several analyses of consumer behavior during the COVID-19 pandemic [52,89] indicate that many changes have occurred. M-commerce applications have offered customers great convenience and comfort in purchasing products during the COVID-19 pandemic, and the correctness of product delivery (meeting the deadline, information on the order delivery status, correct sending and arrival at the destination in good condition of the products ordered) increase customers' confidence in these services [21]. Ease of transaction and customer trust influences their behavioral intention to use mobile commerce [90–93]. Thus, customers' perceptions of the benefits offered by virtual commerce, the post-purchase experience, and how the benefits are perceived compared to other channels positively influence customer attitudes in opting for m-commerce [36,94].

Since the delivery of e-services is critical to successful the consumer adoption of e-commerce, the following hypothesis is suggested:

**Hypothesis 4 (H4).** *Fulfilment (FF) has a positive and significant effect on m-commerce service quality (MQ).*

*2.5. M-Commerce Service Quality and Customer Satisfaction*

Many researchers have studied the concept of e-service quality association with customer satisfaction and repurchase intentions. In the digital age, especially during the pandemic and post-pandemic period, it is essential for companies that want to take advantage of the great opportunities of virtual commerce to provide excellent customer service. At the same time, customer behavior significantly influences the quality of electronic services [40]. It is noted that social influence is a powerful tool for transforming customers' attitudes [41].

Some research has explored the influence of mobile service quality on customer satisfaction and loyalty [8], finding a significant impact of m-commerce service quality on customer satisfaction that, in turn, has a positive impact on loyalty [43].

Consumers have various reasons (trust, social influence, mobility, involvement) for adopting a positive behavior towards m-commerce [85,95]. Thus, the specialized literature has paid particular attention to the examination of factors affecting customer satisfaction in mobile commerce [66,90,96]. Consumer satisfaction is essential to the business process because a satisfied online customer will buy again and recommend it to others [97]. Thus, one of the challenges faced by companies is how to improve online shopper satisfaction [98]. Online customer experience becomes a differentiator for securing a sustainable competitive advantage [99].

Various aspects of customer satisfaction elements (trust, social influence, perceived usefulness and enjoyment, mobility, ease of use, innovation, and engagement) have been analyzed [100]. By carrying out a thorough investigation of the quality of mobile services, the specialized literature confirmed the existence of a significant positive relationship between the quality of electronic services and customer satisfaction [40,69,101].

Thus, the following hypothesis is offered to investigate the effect of service quality on customer satisfaction in online shopping:

**Hypothesis 5 (H5).** *M-commerce service quality (MQ) has a positive and significant effect on Customer satisfaction (CS).*

*2.6. Customer Trust and Loyalty*

The customer's trust in m-commerce is given by the interest that the online store gives him through the quality of the services offered (how it is treated, compliance with

standards, the opinions of other customers), having consequences on the attitude and behavioral intention [40,58].

In addition to changing consumer behavior and perception toward virtual commerce [77], the effects of the COVID-19 pandemic have strengthened consumer confidence in e-commerce through the quality of services obtained [27,53], leading to increased consumer satisfaction in virtual commerce [35]. At the same time, offering guarantees to customers significantly increases their level of the trust in m-commerce applications [29,102,103].

Several studies have identified that consumer trust significantly influences the customer's behavioral intention to use mobile commerce [7,27,40,58,91,98,104,105], due to the quality of services offered [43,102,105,106].

Starting from the studied literature, the following hypothesis is issued:

**Hypothesis 6 (H6).** *M-commerce service quality (MQ) has a positive and significant effect on Customer trust (CT).*

For online retailers, a major challenge is increasing customer loyalty [107], considering their satisfaction as a determining factor [40,43,99,108]. To ensure customer loyalty, suppliers must determine the main factors that influence consumer satisfaction in m-commerce [107].

Repurchase intention indicates the customer's willingness to make another purchase from the same m-commerce application based on the quality of their previous experiences. Some studies have investigated the determinants of repurchase intention and satisfaction in the m-commerce experience [41]. Customer satisfaction increases loyalty to a brand or mobile shopping and prevents them from switching to competitors [101]. Thus, if customers are satisfied with the service provided, they will increase future intentions to use and purchase from the same supplier, so satisfaction influences the intention to repurchase [40,99].

Based on these findings, the following hypothesis is issued:

**Hypothesis 7 (H7).** *Customer satisfaction (CS) has a positive and significant effect on Customer loyalty (CL).*

The literature has shown a continuous interest in research of the factors that influence customer loyalty [82,107]. A success factor to survive in a highly competitive electronic environment is for companies to provide customers with quality service [102] so that they trust and come back to be loyal to the company. When consumers have positive experiences with m-commerce applications, they are very likely to use them again in the future [41,93]. Thus, some studies have identified positive relationships between trust and loyalty [40,102,108]. Customer loyalty leads to an increased customer's number without advertising expenses because loyal customers recommend the supplier to others [109].

Considering the significant effect of trust on consumer loyalty, the following hypothesis is proposed:

**Hypothesis 8 (H8).** *Customer trust (CT) has a positive and significant effect on Customer loyalty (CL).*

Customer satisfaction depends on service quality [66,90,102]. At the same time, it has a positive impact on customer loyalty [41,93], making it a mediating variable between service quality and customer loyalty. Since a positive association was found between e-service quality and customer trust [40,69,101] on the one hand, and between customer trust and loyalty on the other [40,102,108], trust mediates the relationship between service quality and loyalty.

According to these findings, the hypothesis is issued:

**Hypothesis 9 (H9).** *Customer satisfaction (CS) and Customer trust (CT) mediates the relationship between m-commerce service quality (MQ) and Customer loyalty (CL) and at the same time m-commerce service quality (MQ) mediates the relationship between Mobile site design and content (DC), Customer support (CP), Security/Privacy (SP), Fulfillment (FF), and Customer loyalty (CL).*

### 2.7. Conceptual Model Development

In accordance with the reviewed literature [40,43,69,101] and with the formulated hypotheses, we propose a conceptual model with eight builders: (1) Mobile site design and content (DC); (2) Customer support (CP); (3) Security/Privacy (SP); (4) Fulfillment (FF); (5) M-commerce service quality (MQ); (6) Customer satisfaction (CS); (7) Customer trust (CT); and (8) Customer loyalty (CL). The constructs, the number of items for each construct, and their corresponding coding are presented in Table 1. The model variables are divided into four latent exogenous (independent) and four latent endogenous (dependent) variables.

**Table 1.** Constructs, items, and coding for structural model.

| Latent Variables | Construct | Items | Coding | Source |
|---|---|---|---|---|
| Exogenous latent variables | Mobile site design and content (DC) | The content of the mobile site is concise | DC1 | |
| | | The content of the mobile site is accurate | DC2 | |
| | | The mobile site adequately meets my information needs | DC3 | [8,62,67,70–72] |
| | | The mobile site contains all of the content as that on the regular site | DC4 | |
| | | The mobile site contains regularly updated content | DC5 | |
| | | The content provided is fully understandable | DC6 | |
| | | The mobile site displays a visually pleasing design | DC7 | [8,41,67,73] |
| | | The mobile site loads pages quickly | DC8 | |
| | | The mobile site enables me to complete transactions quickly | DC9 | [8,59,67,68] |
| | | It is easy to navigate to any area of the mobile site | DC10 | |
| | | The mobile site has no difficulties with making a payment online | DC11 | [58,70,75–81] |
| | Customer support (CP) | The service agents are able to quickly resolve the problem | CP1 | |
| | | The service agents consistently provide useful advice | CP2 | |
| | | There is a dedicated online chat function on the mobile site | CP3 | |
| | | The mobile site offers the ability to speak to a live person if there is a problem. | CP4 | [30,74,82–85] |
| | | There is a telephone number available to reach the company | CP5 | |
| | | The mobile site has a clear process for handling returns | CP6 | |
| | | The mobile site provides me with convenient options for returning items | CP7 | |
| | | The online shop offers a meaningful guarantee | CP8 | |
| | Security/Privacy (SP) | The mobile site has adequate security features | SP1 | |
| | | This mobile site protects information about my card | SP2 | [36,40,68,69,73, 87,88] |
| | | I trust the mobile site to keep my personal information safe | SP3 | |
| | | It protects information about my mobile-shopping behavior | SP4 | |
| | Fulfilment (FF) | The product is delivered by the time promised by the company | FF1 | |
| | | The mobile site suggests a time frame for when the item will be delivered | FF2 | |
| | | The mobile site sends out the correct items | FF3 | [21,52,89–94] |
| | | The ordered products arrived in good condition | FF4 | |
| | | The mobile site has accurate stock information | FF5 | |
| Endogenous latent variables | M-commerce service quality (MQ) | Overall, my purchase experience with this mobile site is very good | MQ1 | |
| | | The overall quality of the services provided by this mobile site is very good | MQ2 | [40–43] |
| | | My overall feelings toward this mobile site are very satisfied | MQ3 | |
| | Customer satisfaction (CS) | My choice to purchase from the mobile site was wise | CS1 | |
| | | The mobile site has met my expectations | CS2 | [40,66,69,85,90, 95,97–101] |
| | | I did the right thing by choosing this mobile site | CS3 | |
| | | The mobile site enabled a pleasant shopping experience | CS4 | |
| | Customer trust (CT) | This online shop is genuinely interested in customer's welfare | CT1 | [7,27,29,35,40, 43,53,58,77,91, 98,102–106] |
| | | If problems arise, one can expect to be treated fairly by this online shop | CT2 | |
| | | I am happy with the standards by which this online shop is operating | CT3 | |
| | | You can believe the statements of this online shop | CT4 | |
| | Customer loyalty (CL) | I will continue to use the mobile site to shop for new goods | CL1 | [40,41,43,93,99, 101,102,107– 109] |
| | | I would recommend this mobile site to other people | CL2 | |
| | | I will encourage people to purchase from this mobile site | CL3 | |
| | | This mobile site will be my preference when I shop again | CL4 | |

The links between the latent variables within the model are presented in Figure 1.

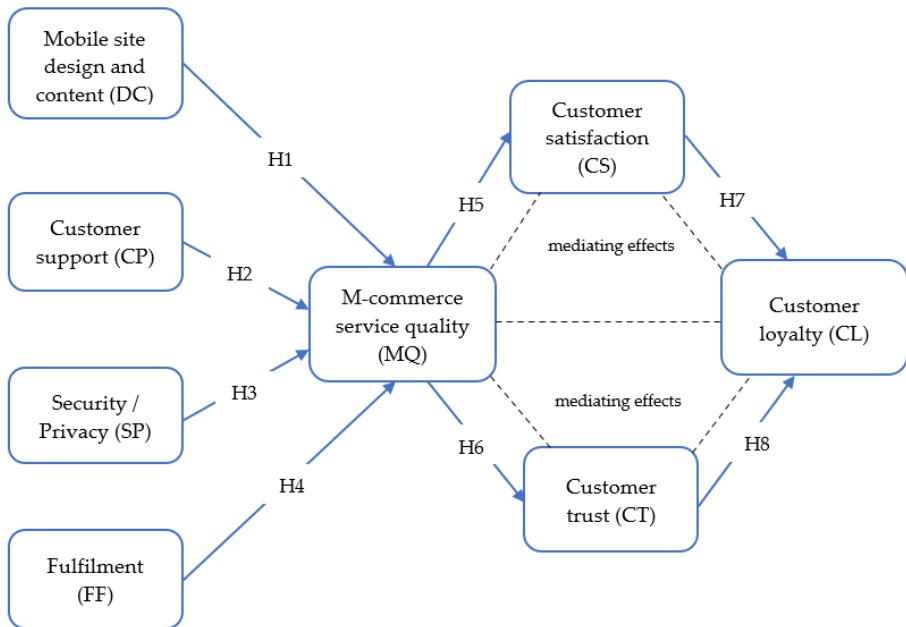

**Figure 1.** Description of the conceptual structural model.

### 3. Methodology

*3.1. Data Collection and Sample Characteristics*

The research analyzes the extent to which the quality of m-commerce services influences consumer behavior. Thus, an empirical survey-based investigation was performed using a 15-question questionnaire. The questionnaire was intended for Romanian m-commerce consumers who are especially active on this trading channel.

At the beginning of 2022, there were 27.41 million cellular mobile connections in Romania, equivalent to 143.7% of the total population. This is because consumers may have more than one mobile connection (a connection for personal use, one for work), so the number of mobile connections significantly exceeds the population. It is worth noting that the number of mobile connections in Romania increased by one million between 2021 and 2022 (+3.9%) [110].

The survey was conducted in 2021 (third and fourth wave of the COVID-19 pandemic) and early 2022 (fifth wave of the COVID-19 pandemic). The sample to whom the survey addressed included only those users who made at least one commercial transaction via mobile phone during 2021–2022, given that in 2021 the majority of the Internet users in Romania used m-commerce applications [111]. Following a study by Zitech regarding the online shopping intention of Romanians in 2022, about 52% of the respondents estimate that they will make as many online purchases as in 2021, and 30% propose to make more online purchases [112].

The questionnaire included two parts. The first part consisted of seven questions regarding the respondents' characteristics (sex, age, education, monthly income, employability, experience in m-commerce, and several transactions in m-commerce) (Table 2). The second part includes eight specific questions identified following the review of specialized literature. Based on the second category of questions, the measurement elements specific to the items and latent variables related to the proposed structural model were constituted (Table 1).

Responses were collected using a Likert scale that provides response categories on a scale from 1 to 5, where the response options were between "not at all = 1", "very little = 2", "average = 3", "a lot = 4" and "very much = 5" [113].

Out of the 1000 questionnaires distributed to the respondents, 738 questionnaires were collected, of which 560 were valid, and the remaining 178 were incomplete or filled in incorrectly, resulting in a response rate of 56%. The margin of error corresponding to a

95% probability of guaranteeing the research results is $+/-3.077\%$, which proves that the sample is representative (margin of error < 5%) [114].

**Table 2.** Sample structure.

| Variables | Category | Number (N) | Percentage (%) |
|---|---|---|---|
| Gender | Female | 327 | 58.39 |
| | Male | 233 | 41.61 |
| Age | 18–25 | 138 | 24.65 |
| | 26–40 | 243 | 43.39 |
| | 41–56 | 117 | 20.89 |
| | >56 | 62 | 11.07 |
| Education | High school diploma | 87 | 15.54 |
| | Bachelor's degree | 189 | 33.75 |
| | Master's degree | 247 | 44.11 |
| | Ph.D. degree and above | 37 | 6.60 |
| Employment | Full time employment | 297 | 53.04 |
| | Part time employment | 16 | 2.88 |
| | Student | 154 | 27.50 |
| | Unemployment | 25 | 4.45 |
| | Retired | 37 | 6.60 |
| | Other | 31 | 5.53 |
| Monthly income | Less than EUR 400 | 112 | 20.00 |
| | EUR 401–1000 | 211 | 37.68 |
| | EUR 1001–1500 | 143 | 25.54 |
| | EUR 1501–2000 | 76 | 13.57 |
| | More than EUR 2001 | 18 | 3.21 |
| M-shopping experience | Less than 12 months | 52 | 9.29 |
| | 1–2 years | 189 | 33.75 |
| | 2–3 years | 163 | 29.11 |
| | More than 3 years | 156 | 27.85 |
| M-shopping: times purchased in 2021–2022 | 1–2 | 22 | 3.93 |
| | 3–4 | 55 | 9.82 |
| | 5–6 | 136 | 24.29 |
| | More than 6 | 347 | 61.96 |
| Total | | 560 | 100 |

An analysis of the respondents' profiles indicates that in the sample there are more women (58.4%) compared to men (41.6%). In terms of age, the sample had the following structure: Generation Z: 18–25 (24.7%), Generation Y: 26–40 (43.4%), Generation X: 41–56 (20.8%), and over 56 years (11.1 %). For education level, respondents with a high school diploma represented 15.5% of the entire sample, followed by those with a university degree (77.9%) and remaining 6.6% of respondents had more than a bachelor's degree. Due to the fact that in Romania there is no official socio-demographic data on the profile of the digital commerce consumer, the structure of the sample is similar to the official government statistics on e-commerce and mobile phone users [47]; thus, it can conclude that the sample represents the analyzed population.

*3.2. SEM–ANN Data Analysis*

This study employs a multi-analytical methodology by integrating Structural Equation Modeling (SEM) with the most powerful artificial intelligence technique, Artificial Neural Network (ANN) [35,38,66]. SEM is an advantageous data analysis method because it simultaneously evaluates measurement and structural models and may oversimplify the complexities of user opinion towards m-commerce platforms' use. To overcome this limitation, we also approached the use of neural networks. The predicting performance of ANN is more powerful than statistical models, such as structural equation modeling and multiple linear regression [115].

To achieve the research objectives, the analysis of the collected data was carried out in two phases, combining structural equation modeling (SEM) with artificial neural network (ANN) analysis by using IBM SPSS Statistics and Amos v.26.0 software [116].

The first phase primarily consisted of processing the collected data in order to verify the correctness of the information provided by the respondents and to eliminate errors, ensuring the interactive validation of the data provided and implicitly of the questionnaire. Then, the reliability of each latent construct in the research model was tested. Secondly, the confirmative factor analysis was used to determine the fit and validity of the proposed model. Thirdly, the research hypotheses were tested using structural equation modeling (SEM) and the proposed structural model was validated.

In the second phase, by performing the neural network analysis, the intensity of the hypothetical relationships previously determined and confirmed by SEM was verified.

### 3.2.1. Structural Equation Modeling

As digital service quality measurement indicators and their influencing factors have abstract and multidimensional characteristics, they are difficult to measure directly. Thus, for the analysis of this complex system, the testing of research hypotheses and the relationships between variables are using SEM [38,55,66]. The proposed structural model consists of four independent variables and four dependent variables (Figure 1). All latent constructs were measured by means of several items based on the review of relevant studies (Table 1).

Using SEM, we are primarily concerned with whether the model produces an estimated population covariance matrix that is consistent with the sampled (observed) covariance matrix. Then, we proceed to the fitting of the model (through the statistics of the chi-square test and goodness-of-fit indices), the reliability of the indicators, the estimation of the parameters for each path in the model (the importance of each path in predicting the measure of the respective result is distinguished, i.e., an independent variable affects a specific dependent variable), and mediating or testing indirect effects (the independent variable affects the dependent variable through a mediating variable) [117].

$$x = \Lambda_x \xi + \delta \tag{1}$$

$$y = \Lambda_y \eta + \varepsilon \tag{2}$$

The Equations (1) and (2) stipulate the relationship between the result latent variable $\eta$ and the result observable variable $y$, and the relationship between the cause latent variable $\xi$ and the cause observable variable $x$. $\Lambda_x$ is the relationship between the cause observable variable and the cause latent variable, and is the factor loading matrix of the cause observable variable on the cause latent variable. $\Lambda_y$ is the relationship between the result observable variable and the result latent variable, and is the factor loading matrix of the result observable variable on the result latent variable. $\delta$ is the error of the cause observable variable $x$; $\varepsilon$ is the error of the result observable variable $y$. The structural model equation is set as follows:

$$\eta = \beta\eta + \Gamma\xi + \zeta, \tag{3}$$

where $\beta$ is the coefficient matrix of the result latent variable $\eta$, and also the path coefficient matrix between the result latent variables; $\Gamma$ is the coefficient matrix of the cause latent variable $\xi$, and also the path coefficient matrix of the cause latent variable to the corresponding endogenous latent variable; $\zeta$ is the residual term of the structural equation, which is the part failed to explain within the model.

### 3.2.2. ANN

ANN is applied to examine the complement and verify the SEM analysis and measure the effectiveness of independent factors on the dependent factor. From the category of artificial intelligence techniques, ANN is widely used, due to its efficiency in modeling complex relationships between inputs and outputs. ANN has higher prediction

accuracy compared to SEM and does not require the formulation of hypotheses to be tested [35,38,66,118].

The type of neural network used was the multilayer perceptron model (MLP) because it is very flexible to be applied to the respective indicators analyzed in the paper and to learn the mapping from inputs to outputs and is compatible with regressions identified as the most appropriate for the data used. The MLP neural network model has an architecture of three main layers: an input, output, or hidden layer. For each neural network built, the activation functions justify: (1) for the hidden layer, the capacity and performance of the neural network; (2) for the output layer, the validity of the chosen regression model. The synaptic weight establishes for each neural network the amplitude of the connection between the nodes and the meaning of the relationship (direct or indirect) between the nodes. Using the additional parameter bias helps the built neural network architecture to best fit the analyzed data [119].

## 4. Empirical Findings and Discussion

### 4.1. Reliability and Validity Analysis

Before the actual analysis of the model, the degree of significance of the variables of the structural conceptual model is checked, namely, the reliability, validity, and internal consistency of the collected data are analyzed (Table 3).

**Table 3.** The measurement model results.

| Construct | CA | CR | DG rho | AVE | SR AVE | VIF |
|---|---|---|---|---|---|---|
| Mobile site design and content (DC) | 0.922 | 0.894 | 0.915 | 0.713 | 0.844 | 1.490 |
| Customer support (CP) | 0.886 | 0.866 | 0.904 | 0.656 | 0.810 | 1.150 |
| Security/Privacy (SP) | 0.919 | 0.774 | 0.778 | 0.740 | 0.860 | 1.720 |
| Fulfilment (FF) | 0.866 | 0.806 | 0.849 | 0.699 | 0.836 | 1.910 |
| M-commerce service quality (MQ) | 0.826 | 0.706 | 0.754 | 0.643 | 0.802 | 3.126 |
| Customer satisfaction (CS) | 0.824 | 0.764 | 0.808 | 0.657 | 0.810 | 4.322 |
| Customer trust (CT) | 0.773 | 0.760 | 0.826 | 0.628 | 0.793 | 3.885 |
| Customer loyalty (CL) | 0.837 | 0.763 | 0.810 | 0.654 | 0.809 | |

Notes: Cronbach's Alpha (CA); Composite Reliability (CR); Dillon Golstein's rho (DG rho); Average Variance Extracted (AVE); Square root of AVE (SR AVE); Variance Inflation Factor (VIF).

First of all, the reliability of the internal consistency of the questionnaire was verified. For this, it was estimated the internal consistency of the items that were used to measure the latent variables by calculating Cronbach's Alpha, composite reliability, and Dillon Golstein's rho. In all of the cases, CA and CR values are greater than the threshold of 0.7 and AVE exceeded the critical value of 0.5, which means that the reliability criteria was met; that is, the exogenous variables in the model are statistically significant [120–122]. Very high reliability was obtained for most of the variables ((CA > 0.9: DC (0.922) and SP (0.919), or great (0.9 < CA > 0.8: CP (0.886); FF (0.866); MQ (0.826); and CS (0.824)).

Table 3 shows that there are no multicollinearity problems between exogenous variables in the model, because the values of the variation inflation factor (VIF) do not exceed the value of 5 [123].

In the next stage, the validity of the proposed model was tested and the fit was determined using confirmative factor analysis. Validity measures whether the variables in the model can really be expressed by the appropriate measurement elements.

Kaiser–Meyer–Olkin (KMO) Test and Bartlett's Test of Sphericity are used for validity analysis. The validity of the model is higher as the measurement results are more consistent with the content to be measured. KMO Test is used to compare the relative size of the Pearson correlation coefficient and that of the original variables. The KMO value is between 0 and 1.

The correlation between the variables is stronger and more suitable for factor analysis as the value is closer to 1 [124]. Bartlett's sphericity test is used to test whether the correlation matrix is an identity matrix, that is, whether each variable is independent, indicating that the data are not suitable for factor analysis. However, the lower the level of significance, the higher the likelihood of a significant relationship between the original variables. The test results of this study are shown in Table 4.

**Table 4.** KMO and Bartlett's Test.

| Kaiser–Meyer–Olkin Test of Sampling Adequacy | | 0.803 |
|---|---|---|
| Bartlett's Test of Sphericity | Approx. chi-square | 160.790 |
| | df | 68 |
| | Sig. | 0.000 |

In Table 4, the KMO value is 0.803, indicating that the factor analysis effect is good. The value of Bartlett's test of sphericity is 160.790. When the degree of freedom is 68, $p < 0.05$, reaching the significance level. Therefore, this study is suitable for factor analysis.

The Average Variance Extracted (AVE) indicator is used to test the discriminant validity of the model. Thus, the square root of the AVE in each construct is compared with its inter-construct correlation for all latent variables in the model and is observed to be higher, thus confirming the discriminant validity of the proposed model (Table 5) [125].

**Table 5.** Fornell–Larcker Criterion Analysis for Discriminant Validity.

| | DC | CP | SP | FF | MQ | CS | CT | CL |
|---|---|---|---|---|---|---|---|---|
| DC | 0.844 | | | | | | | |
| CP | 0.071 | 0.810 | | | | | | |
| SP | 0.241 | 0.115 | 0.860 | | | | | |
| FF | 0.376 | 0.169 | 0.399 | 0.836 | | | | |
| MQ | 0.612 | 0.207 | 0.739 | 0.718 | 0.802 | | | |
| CS | 0.450 | 0.387 | 0.415 | 0.637 | 0.776 | 0.810 | | |
| CT | 0.344 | 0.344 | 0.529 | 0.643 | 0.675 | 0.791 | 0.793 | |
| CL | 0.472 | 0.243 | 0.504 | 0.694 | 0.791 | 0.801 | 0.790 | 0.809 |

*4.2. Model Fitting*

The fitting effect of structural equation model indicates whether the interaction between variables exists. By testing the fitting effect of the model, the model is continuously optimized until the model with the best fitting effect is found. The used goodness-of-fit indices of structural equation model are: Ratio of chi-square to degree of freedom ($\frac{\chi^2}{df}$); Goodness-of-Fit Index (GFI); Standardized root mean square residual (SRMR); Root Mean Square Error of Approximation (RMSEA); Normed Fit Index (NFI); Incremental Fit Index (IFI); and Comparative Fit Index (CFI).

After testing the model, it is found that it is necessary to exclude the factor "The mobile site contains all of the content as that on the regular site" (DC4) with the value of 0.417, because it does not meet the ideal standard of 0.5 [126].

The Modification Indices (MI) option in AMOS 26.0 software is used to find the path with the maximum MI value and add the path, that is, the initial model is modified once; then is running the AMOS 26.0 software to obtain the goodness-of-fit index value of the modified model to verify the fitting effect of the modified model.

The above steps are repeated until the value of the goodness-of-fit index of the modified model approaches or meets the requirements of the standard value.

According to the above method, the initial model is modified several times, and finally the optimal structural equation model is obtained, as shown in Figure 2.

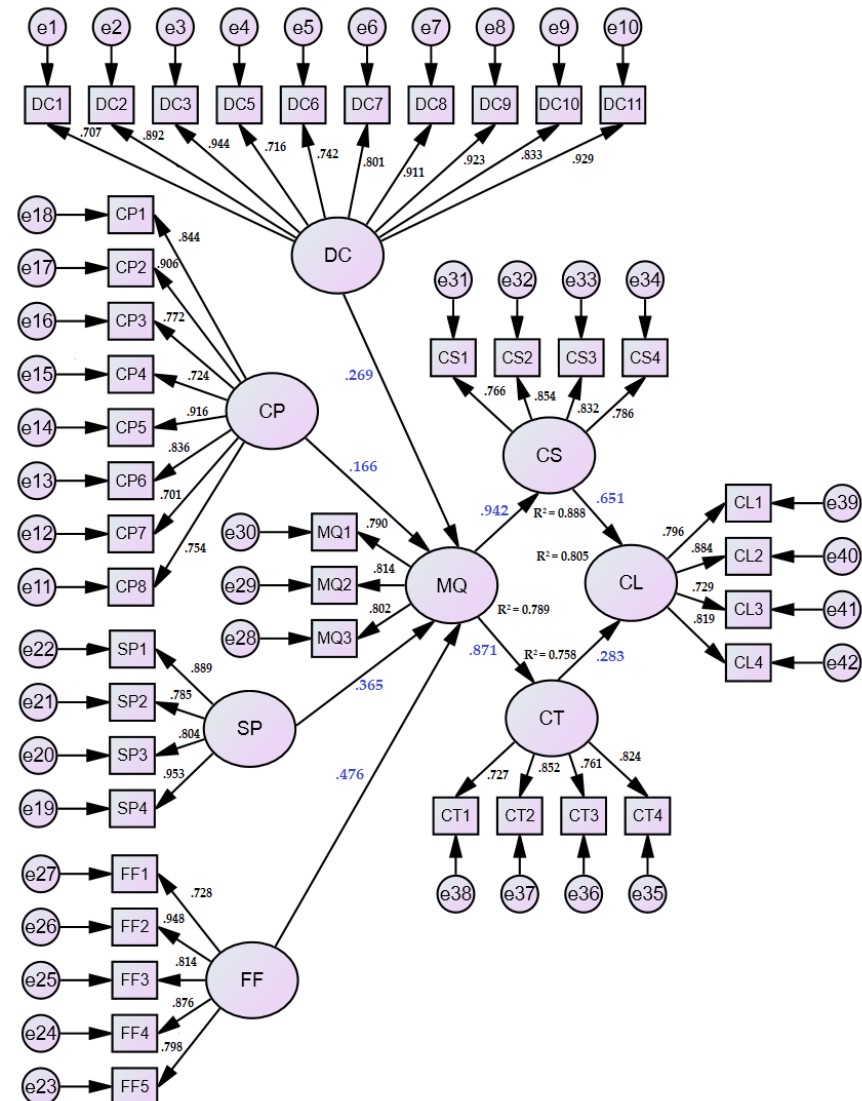

**Figure 2.** SEM measurement model. Source: developed using SPSS Amos v.26.0.

The values of the goodness-of-fit indices of the modified structural equation model are shown in Table 6.

**Table 6.** Fit summary of criteria and initial model.

| Fit Indices | $\frac{\chi^2}{df}$ | GFI | SRMR | RMSEA | NFI | IFI | CFI |
|---|---|---|---|---|---|---|---|
| Recommended value | <2 | >0.90 | <0.08 | <0.06 | >0.90 | >0.90 | >0.90 |
| Source | [127] | [128] | [129] | [129] | [130] | [130] | [131] |
| Modified Model | 1.468 | 0.925 | 0.073 | 0.058 | 0.916 | 0.952 | 0.947 |

The most of the fit indices of the modified model have been improved. The fit degree of the model is relatively good and basically passes the test of goodness-of-fit. In other words, the correlation between variables in the modified structural equation model shown in Figure 2 does exist.

*4.3. Analysis of Direct and Mediation Effects*

The path coefficient and hypothesis testing results of the theoretical model are shown in Table 7.

**Table 7.** Hypothesis (H) testing results.

| Hypothesis | Paths Correlation | Path Coeff. (β) | *p* | Results |
|---|---|---|---|---|
| H1 | Mobile site design and content (DC) → M-commerce service quality (MQ) | 0.269 | *** | Supported |
| H2 | Customer support (CP) → M-commerce service quality (MQ) | 0.166 | *** | Supported |
| H3 | Security/Privacy (SP) → M-commerce service quality (MQ) | 0.365 | *** | Supported |
| H4 | Fulfilment (FF) → M-commerce service quality (MQ) | 0.476 | *** | Supported |
| H5 | M-commerce service quality (MQ) → Customer satisfaction (CS) | 0.942 | *** | Supported |
| H6 | M-commerce service quality (MQ) → Customer trust (CT) | 0.871 | *** | Supported |
| H7 | Customer satisfaction (CS) → Customer loyalty (CL) | 0.651 | *** | Supported |
| H8 | Customer trust (CT) → Customer loyalty (CL) | 0.283 | *** | Supported |

Note: *** $p < 0.05$. Covariances: (Group number 1—Default model).

According to the data in Table 7, the *p* values of the expected hypotheses H1-H8 are all less than 0.05, indicating that these hypotheses are supported by data.

Thus, we will analyze the direct effects between MQ and its predictors (DC, CP, SP, and FF), MQ and its successors CS and CT; CL and its predictors CS and CT.

From the analysis of the direct results, it is observed that DC has a positive and significant impact on the MQ ($\beta = 0.269$; $p < 0.05$). This result indicates that an improvement of m-commerce applications (content, design, security, ease of use), lead to an increase in the quality of services offered by these solutions. It is found that the impact of CP on MQ ($\beta = 0.166$; $p < 0.05$) is positive and statistically significant, which supports the positive role of customer support on the quality of m-commerce services.

At the same time, we observe a positive and statistically significant coefficient between SP and MQ ($\beta = 0.365$; $p < 0.05$), which suggests that security and information protection have a very important role in terms of the quality of services offered by m-commerce.

Similarly, it is found that FF has a significant and positive impact on MQ ($\beta = 0.476$; $p < 0.05$), which demonstrates the importance of optimizing the transaction process on the quality of the services offered by m-commerce.

Considering that MQ has a significant and direct impact on CS ($\beta = 0.942$; $p < 0.05$), we can say that the quality of m-commerce services is a decisive factor in customer satisfaction. Similarly, it turns out that MQ has a significant and positive impact on CT as well ($\beta = 0.871$; $p < 0.05$), which highlights the role that the quality of m-commerce services has in increasing customer trust. We observe that the impact of CS on CL ($\beta = 0.651$; $p < 0.05$) is positive and statistically significant, which supports the role that customer satisfaction has on their loyalty. From the analysis of the direct results, it follows that CT has a positive and significant impact on CL ($\beta = 0.283$; $p < 0.05$). This result indicates that customer trust influences their loyalty.

In addition to the analyzed direct relationships, in the proposed model, we identified and analyzed the mediating effect of MQ on the relationship between DC, CP, SP, FF and CS, CT, CL and the mediating effect of CS and CT, respectively, on the relationship between MQ and CL. Table 8 shows all the specific standardized effects (direct according to Table 7, indirect and total) and it is noted that the *p*-values of the structural relationships are all less than 0.05.

From the analysis of the results in Table 8, the following can be observed:

- MQ has a mediating effect on CS through the four exogenous latent variables: DC ($\beta = 0.254$; $p < 0.05$), CP ($\beta = 0.156$; $p < 0.05$), SP ($\beta = 0.344$; $p < 0.05$), and FF ($\beta = 0.448$; $p < 0.05$);
- MQ has a mediating effect on CT through the four exogenous latent variables DC ($\beta = 0.234$; $p < 0.05$), CP ($\beta = 0.144$; $p < 0.05$), SP ($\beta = 0.318$; $p < 0.05$), and FF ($\beta = 0.414$; $p < 0.05$);
- MQ has a mediating effect on CL through the four exogenous latent variables DC ($\beta = 0.231$; $p < 0.05$), CP ($\beta = 0.142$; $p < 0.05$), SP ($\beta = 0.313$; $p < 0.05$), and FF ($\beta = 0.409$; $p < 0.05$), which generates an indirect positive impact on CS, CT, and CL;
- CS and CT, respectively, mediate the effect between MQ and CL ($\beta = 0.859$; $p < 0.05$).

**Table 8.** The standardized total, direct, and indirect effects registered between the variables of the structural equations modeling.

|  | DC | CP | SP | FF | MQ | CS | CT |
|---|---|---|---|---|---|---|---|
|  | Standardized total effects | | | | | | |
| MQ | 0.269 | 0.166 | 0.365 | 0.476 | 0.000 | 0.000 | 0.000 |
| CS | 0.254 | 0.156 | 0.344 | 0.448 | 0.942 | 0.000 | 0.000 |
| CT | 0.234 | 0.144 | 0.318 | 0.414 | 0.871 | 0.000 | 0.000 |
| CL | 0.231 | 0.142 | 0.313 | 0.409 | 0.859 | 0.651 | 0.283 |
|  | Standardized direct effects | | | | | | |
| MQ | 0.269 | 0.166 | 0.365 | 0.476 | 0.000 | 0.000 | 0.000 |
| CS | 0.000 | 0.000 | 0.000 | 0.000 | 0.942 | 0.000 | 0.000 |
| CT | 0.000 | 0.000 | 0.000 | 0.000 | 0.871 | 0.000 | 0.000 |
| CL | 0.000 | 0.000 | 0.000 | 0.000 | 0.000 | 0.651 | 0.283 |
|  | Standardized indirect effects | | | | | | |
| MQ | 0.000 | 0.000 | 0.000 | 0.000 | 0.000 | 0.000 | 0.000 |
| CS | 0.254 | 0.156 | 0.344 | 0.448 | 0.000 | 0.000 | 0.000 |
| CT | 0.234 | 0.144 | 0.318 | 0.414 | 0.000 | 0.000 | 0.000 |
| CL | 0.231 | 0.142 | 0.313 | 0.409 | 0.859 | 0.000 | 0.000 |

*4.4. Artificial Neural Network Analyses*

As discussed in the methodology section, Artificial Neural Network analysis is used in the second phase of the analysis. Significant hypothesized predictors are utilized as inputs to ANN to emphasize the relevant importance of each predictor's variable. ANN produces more precise predictions compared to the SEM approaches [132].

For this study, the independent variables (DC, CP, SP, FF, MQ, CS, and CT) were the input layers (input neurons) of the ANN model. This means the number of input neurons (input layers) equal seven, which is the number of predictors. In construct, the output layer in the ANN model was the dependent variable, which is CL, as shown in Figure 3. This means the number of output neurons (output layers) equals one. Following recommendations given by [133], the hidden neurons (nodes) are automatically generated and activation function (Sigmoid Function) is utilized for both hidden and output layers. Furthermore, based on the recommendations from the above authors, the prediction accuracy of the trained network is measured using ten-fold cross-validation. To avoid overfitting, problem data is divided into two parts, from which 80% for training and 20% for testing.

The prediction accuracy of the ANN model was computed by the root mean square error (RMSE) for both the training (80%) and testing (20%) data sets (ten runs). The RMSE is calculated using Equation (4) [134], where SSE is the sum of squared error, and n is number of items.

$$\text{RMSE} = \sqrt{\frac{1}{n} \times \text{SSE}} \qquad (4)$$

As shown in Table 9, the RMSE values for the training data set and the testing data set represent an accurate ANN model in taking the relationships between predictors and the output. According to [135] lower RMSE values represent higher predictive accuracy and better data fit.

The relative importance of each input predictor was computed in terms of normalized relative importance ranking (expressed as %) using sensitivity analysis as presented in Table 10 and Figure 4.

The results in Table 10 and Figure 4 show that all seven independent variables are important to all 10 neural networks. Based on the normalized variable importance, customer satisfaction is the most significant predictor of customer loyalty followed by customer trust, while customer support has a weaker influence followed by mobile site design and content.

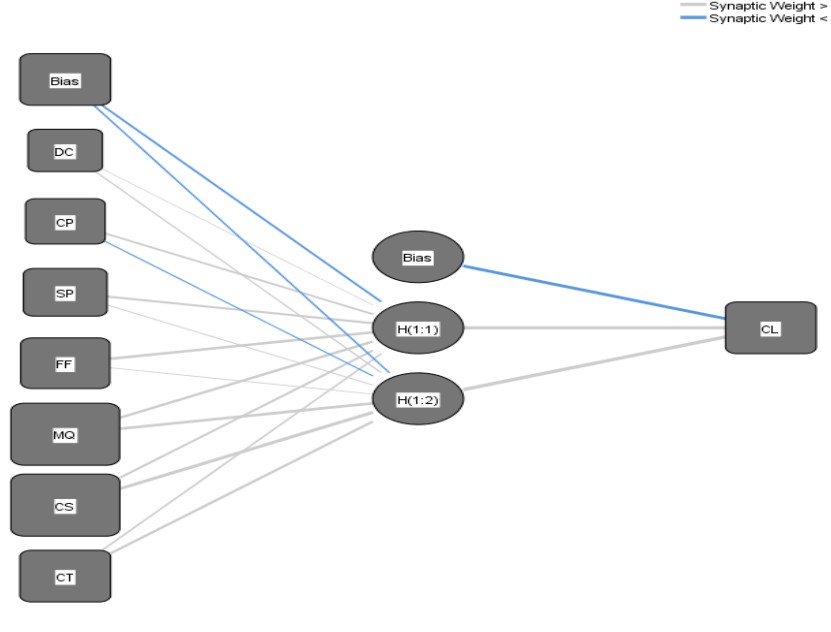

**Figure 3.** Neural Network Model. Source: Developed by the authors based on the data calculated with SPSS Statistics.

**Table 9.** RMSE values for the ANN model.

| Neural Network | Input: DC, CP, SP, FF, MQ, CS, CT Output: CL | | | |
| | Training Dataset (80% of Data Sample 560, n = 448) | | Testing Dataset (20% of Data Sample 560, n = 112) | |
| | SSE | RMSE | SSE | RMSE |
|---|---|---|---|---|
| ANN1 | 0.138 | 0.0176 | 0.127 | 0.0337 |
| ANN2 | 0.133 | 0.0172 | 0.124 | 0.0333 |
| ANN3 | 0.129 | 0.0170 | 0.119 | 0.0326 |
| ANN4 | 0.126 | 0.0168 | 0.113 | 0.0318 |
| ANN5 | 0.131 | 0.0171 | 0.130 | 0.0341 |
| ANN6 | 0.127 | 0.0168 | 0.132 | 0.0343 |
| ANN7 | 0.119 | 0.0163 | 0.126 | 0.0335 |
| ANN8 | 0.118 | 0.0162 | 0.132 | 0.0343 |
| ANN9 | 0.116 | 0.0161 | 0.127 | 0.0337 |
| ANN10 | 0.110 | 0.0157 | 0.104 | 0.0305 |
| | **Mean** | **0.0167** | **Mean** | **0.0332** |

**Table 10.** Normalized variable relative importance.

| Predictors (Independent Variables) | Average Relative Importance | Normalized Importance (%) | Ranking |
|---|---|---|---|
| DC | 0.154 | 46.2 | 7 |
| CP | 0.178 | 53.5 | 6 |
| SP | 0.185 | 55.6 | 5 |
| FF | 0.258 | 77.5 | 4 |
| MQ | 0.333 | 100 | 1 |
| CS | 0.306 | 91.9 | 2 |
| CT | 0.286 | 85.9 | 3 |

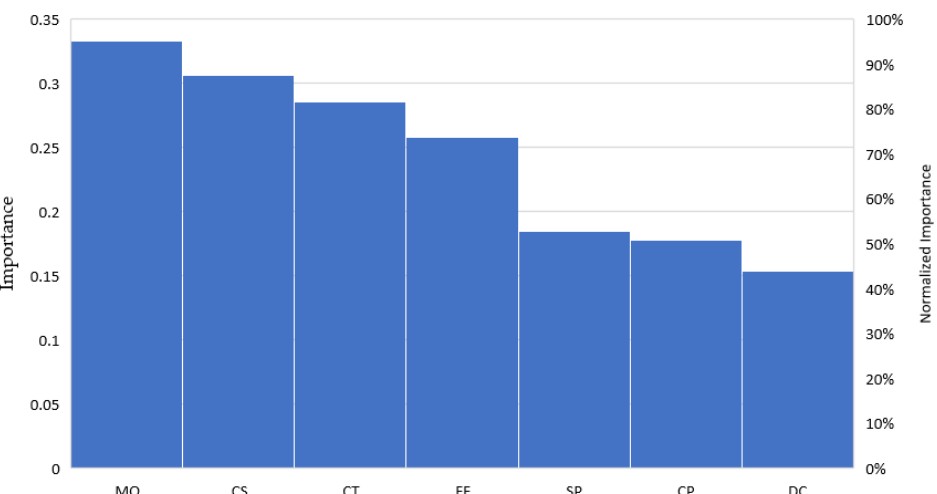

**Figure 4.** Normalized importance. Source: Developed by the authors based on the data calculated with SPSS Statistics.

## 5. Conclusions

The COVID-19 epidemic has changed where and how people buy goods and has accelerated structural changes in consumer preferences, which have affected consumption channels, how retailers interact with each other (business-to-business relationships), and how firms work with their direct suppliers, wholesalers, and distributors. The pandemic has disrupted the value chain and changed consumption patterns. With the appearance of the first cases of COVID-19 worldwide, the demand for processed/canned foods and pharmaceuticals increased substantially. In an environment where habits and practices have changed so quickly and are likely to continue to change, sales leaders need a clear view of what customers want and what steps their company can take to meet their needs. Traditional commerce (physical presence in the store without the need for the Internet) will remain essential in the future, but it will be possible to combine it through mobile applications with a mobile phone by scanning QR codes on product labels. The COVID-19 pandemic came to the aid of consumers by implementing various digital innovations at the level of companies but also at the level of communities to increase the quality of life and achieve sustainable development [136,137].

### 5.1. Theoretical Implications

The COVID-19 pandemic has affected the lives of consumers and their shopping patterns, so more and more people are choosing to do their shopping through m-commerce applications. This research was designed to investigate the impact of the quality of electronic services on the behavior of Romanian consumers under the conditions of the COVID-19 pandemic. The first objective of this study was to identify, for the period of the COVID-19 pandemic, the significant factors that can lead to an increase in the quality of electronic services offered to Romanian m-commerce customers to ensure loyalty and continuity of transactions on the mobile sales channel. Then, based on the specialized literature, we identified the most significant predictors of the quality of services offered to m-commerce consumers and developed a conceptual model for evaluating the impact of the quality of electronic services on the behavior of m-commerce consumers.

The research is based on the analysis of data collected following the empirical investigation carried out in 2021–2022, which included three waves of the COVID-19 pandemic, on a sample of 560 people. To carry out the research, we used a hybrid SEM–ANN approach, which is consistent with other studies [35,38,66,118]. After previously checking that the variables of the proposed structural model are reliable and valid, we created the most suitable model.

The results of hypothesis testing show consistency and convergence in assessing the extent to which the eight identified factors reinforce and contribute to a clearer understanding of the connections within the proposed model. The empirical results demonstrated that the impact of Mobile site design and content, Customer support, Security/Privacy, and Fulfillment on M-commerce service quality is positive and statistically significant, which supports the importance of implementing efficient, attractive, and secure m-commerce solutions in the conditions of vulnerability generated by the pandemic. These results demonstrate the validity of hypotheses H1–H4 and are supported by the findings of other studies [40–42,52,59,67,70–72,76,80].

At the same time, the results identify the significant role of m-commerce service quality, especially in the COVID-19 crisis, on consumer satisfaction and trust, demonstrating the validity of Hypotheses 5 and 6, being consistent with the results of other specialized studies [27,40,42,43,53,69,101,102,105,106].

The findings indicate that consumer satisfaction and trust have a significant impact on building their loyalty, which validates Hypotheses 7 and 8 and confirms the results of other studies [40,41,93,99,101]. Based on these findings, we can say that if customers are satisfied and trust the service offered, they will return to make new purchases.

Based on the analysis of direct and indirect effects, it is found that the most significant effect on loyalty is exerted by service quality (indirect effect), followed by satisfaction and trust (direct effects). The analysis of mediation effects confirms that in the optimized model (through indirect effects), m-commerce service quality, mobile site design and content, customer support, security/privacy, and fulfillment have a positive and significant impact on loyalty.

The results from both SEM and ANN show that satisfaction and trust are the most important factors for increasing consumer loyalty in virtual commerce, which is consistent with the findings of other researchers [35,38,66].

From the customers' point of view, mobile applications used for business transactions provide them with a convenient way to search, order, locate, or transact through their smartphones anywhere and at any time.

This research contributes to the enrichment of specialized literature through the originality of the structural model that allowed the examination of the quality dimensions of m-commerce services and their impact on customer satisfaction and loyalty in the context of the COVID-19 pandemic. The research findings are robust and support the stability of the proposed conceptual model.

An important study result is identifying the degree to which the factors of the structural model compete to increase the quality of electronic services and thus companies can act punctually on their improvement. At the same time, the research results emphasize the need to increase the quality of services offered by m-commerce applications and ensure adequate security and privacy to increase customer trust in these applications and implicitly the satisfaction that leads to customer loyalty for a particular online store.

### 5.2. Practical Implications

In addition to theoretical implications, our study also reveals some practical implications. The study results provide a clear picture for companies that adopt this sales channel. Thus, to create and increase loyalty in m-commerce applications, it is necessary to provide quality services that ensure a high degree of security and confidentiality. The COVID-19 pandemic has led to a further increase in the popularity of mobile commerce, which challenges companies to constantly adapt to the needs of consumers, highlighting that businesses need fast and efficient ways to serve and communicate with their customers. The online market is complex, very dynamic, and competitive. In this sense, m-commerce offers new opportunities for companies, where they can increase the competitive advantages of the business. The findings of this study can provide managers with insight to better understand the factors that influence e-service quality and the importance of each to ensuring customer satisfaction and trust, which can lead to gaining customer loyalty.

Thus, they can improve their online store service quality and make future strategies by combining with market trends.

Following the results of the research, companies that carry out transactions on the m-commerce channel can consider that to attract and maintain their customers, they must continuously develop the functionality of the application, have an attractive, easy design, offer advanced filtering tools and refinement while browsing and searching, and offer customers personalized offers to convince them to buy through the app and develop brand loyalty. At the same time, this study highlights that positive consumer attitudes towards m-commerce mobile applications lead to favorable brand behavioral outcomes and increased loyalty.

*5.3. Limitations and Future Work*

In addition to theoretical and practical contributions, this study also has limitations, which are worth studying and may suggest future research directions. First of all, this study has a sample formed only of consumers from Romania, which limits the generalization of the results obtained because the specific characteristics of the country and the culture, and the degree of digitalization can influence the quality of m-commerce services and implicitly the consumer behavior. Therefore, future research should conduct cross-country comparative analyzes to test the universality of the proposed model and verify whether this study's findings hold for other countries.

Other limitations are determined by the way in which samples of respondents are formed and the type of information taken from the questionnaires. Future research could use smart algorithms that study e-market customers in order to assess the age, experience, and specific factors and to see if there are some differences between consumer samples that can be classified with respect to these characteristics.

Future research is intended to remove these limitations by combining the use of quantitative and qualitative methods. Potential future research directions could extend the analysis conducted in this study by using a quantitative and qualitative comparative analysis on clusters.

**Author Contributions:** Conceptualization, A.M. and G.S.; formal analysis, A.M. and G.S.; investigation, A.M. and G.S.; methodology, A.M. and G.S.; resources, A.M. and G.S.; software, A.M. and G.S.; validation, A.M. and G.S.; writing—original draft, A.M. and G.S. All authors have read and agreed to the published version of the manuscript.

**Funding:** This research received no external funding.

**Data Availability Statement:** Not applicable.

**Acknowledgments:** The authors acknowledge the anonymous reviewers whose suggestions and comments helped improving the paper.

**Conflicts of Interest:** The authors declare no conflict of interest.

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
