# Peer review of "A Hybrid SEM-Neural Network Modeling of Quality of M-Commerce Services under the Impact of the COVID-19 Pandemic"

_electronics, doi:10.3390/electronics11162499_

Round 1
Reviewer 1 Report
As much as I would instinctively see this type of article in another magazine more focused on commerce than on electronics per se, the article comes across as well-written and well-structured. The purpose of the research is clear as is the methodology used. The results are well presented within tables and figures that are fit for purpose.
Not being an econometrician, I hope others will go into the detail of the models.
I suggest going into some more detail.
- in the introduction, the pre- vs. post-pandemic situation in general and provide data on m shares vs. e- commerce
- I would make some points (in the conclusions maybe...?) about the relationship between product type purchased and device used - thinking of drugs and grocery especially, which may be handled differently. On grocery, which I know best, and on which there is much in the literature, see, in particular:
Dannenberg, P., Fuchs, M., Riedler, T., & Wiedemann, C. (2020). Digital transition by COVID‐19 pandemic? The German food online retail. Tijdschrift voor economische en sociale geografie, 111(3), 543-560.
Asti, W. P., Handayani, P. W., & Azzahro, F. (2021). Influence of trust, perceived value, and attitude on customers’ repurchase intention for e-grocery. Journal of Food Products Marketing, 27(3), 157-171.
Sernicola F., Maltese I., Gatta V., Iannaccone G., Marcucci E. (2020), Impatto del lockdown sulla spesa degli italiani: quale futuro per l’e-grocery?, REPoT-Rivista di Economia e Politica dei Trasporti, n.3, art.4, pp.1-13https://www.openstarts.units.it/handle/10077/32169
Grashuis, J., Skevas, T., & Segovia, M. S. (2020). Grocery shopping preferences during the COVID-19 pandemic. Sustainability, 12(13), 5369.
Ryadi, W. T., Kurniasari, F., & Sudiyono, K. A. (2021). Factors influencing consumer's intention towards e-grocery shopping: An extended technology acceptance model approach. Economics, Management and Sustainability, 6(2), 146-159.
Minor point: lines 48-50
"The benefits of m-commerce (availability, purchasing facilities in terms of time and space) give consumers a more user friendly experience than e-commerce". PLS provide a source.
Author Response
The comments of the reviewer are taken into consideration and answered in attached letter.
The authors believe that the revised version of the paper is better than the originally submitted one. For this reason, they reiterate their gratitude to the anonymous reviewer.

Reviewer 2 Report
The reviewed article in general must be evaluated more than positively. explores a topic that is relevant not only in the context of the COVID-19 pandemic but also for the future development of trading. The electronic form of trading is more than promising, it will probably eventually replace more than 2/3 of brick-and-mortar stores.
Despite the high-quality processing, the authors neglected to focus on the very definition of the term consumer and the term e-commerce. It is customary that non-lawyer authors do not consider it necessary to focus on the definition of these terms. And precisely the correct and unambiguous definition of terms is always the key to achieving research success. It is necessary to clearly know which group of people the research is focusing on.
In this context, I point to several foreign works that deal either directly or indirectly with this issue and whose works you have improved the scientific article under review. It is about:
Peráček, T. (2022). E-commerce and its limits in the context of consumer protection: the case of the Slovak Republic. Juridical Tribune - Tribuna Juridica, 12 (1), pp. 35-50, doi: 10.24818/TBJ/2022/12/1.03
In the context of bringing public services closer to the citizen also within the concept of smart cities, and to increase the citizens' quality of life, even under the pressure of circumstances such as the Covid-19 pandemic brought, also while respecting the principles of sustainable development and, to improve the efficiency and quality of the public services provided, many cities and municipalities strive, for example, to bring various digital innovations as an option digital payments via QR codes. It is, therefore, necessary in this research problem, to take into account such research
Vladimíra Žofčinová, Čajková, Andrea and Král Rastislav. "Local Leader and the Labor Law Position in the Context of the Smart City Concept through the Optics of the EU" TalTech Journal of European Studies, vol.11, no.2, 2021, pp.3-26. doi: 10.2478/bjes-2022-0001
Charaia, V; Chochia, A and Lashkhi, M. (2021). PROMOTING FINTECH FINANCING FOR SME IN S. CAUCASIAN AND BALTIC STATES, DURING THE COVID-19 GLOBAL PANDEMIC. BUSINESS MANAGEMENT AND ECONOMICS ENGINEERING 19 (2), pp.358-372, doi: 10.3846/bmee.2021.14755
Author Response

(The authors gave the same response as above.)

Round 2
Reviewer 2 Report
I am glad that the authors accepted my comments, on the basis of which I recommend the article for publication.